# Unraveling the Roles of miR-204-5p and HMGA2 in Papillary Thyroid Cancer Tumorigenesis

**DOI:** 10.3390/ijms241310764

**Published:** 2023-06-28

**Authors:** Cindy Van Branteghem, Alice Augenlicht, Pieter Demetter, Ligia Craciun, Carine Maenhaut

**Affiliations:** 1Institut de Recherche Interdisciplinaire en Biologie Humaine et Moléculaire (IRIBHM), Université libre de Bruxelles, 1070 Brussels, Belgium; cindy.van.branteghem@ulb.be (C.V.B.); alice.augenlicht@gmail.com (A.A.); 2Anatomie Pathologique, Hôpital Universitaire de Bruxelles, Université libre de Bruxelles, 1070 Brussels, Belgium; pieter.demetter@hubruxelles.be (P.D.); ligia.craciun@hubruxelles.be (L.C.)

**Keywords:** thyroid cancer, papillary thyroid carcinoma, microRNAs, miR-204-5p, HMGA2, MAPK, EMT

## Abstract

Thyroid cancer is the most common endocrine malignant tumor with an increasing incidence rate. Although differentiated types of thyroid cancer generally present good clinical outcomes, some dedifferentiate into aggressive and lethal forms. However, the molecular mechanisms governing aggressiveness and dedifferentiation are still poorly understood. Aberrant expression of miRNAs is often correlated to tumor development, and miR-204-5p has previously been identified in papillary thyroid carcinoma as downregulated and associated with aggressiveness. This study aimed to explore its role in thyroid tumorigenesis. To address this, gain-of-function experiments were performed by transiently transfecting miR-204-5p in thyroid cancer cell lines. Then, the clinical relevance of our data was evaluated in vivo. We prove that this miRNA inhibits cell invasion by regulating several targets associated with an epithelial-mesenchymal transition, such as SNAI2, TGFBR2, SOX4 and HMGA2. HMGA2 expression is regulated by the MAPK pathway but not by the PI3K, IGF1R or TGFβ pathways, and the inhibition of cell invasion by miR-204-5p involves direct binding and repression of HMGA2. Finally, we confirmed in vivo the relationship between miR-204-5p and HMGA2 in human PTC and a corresponding mouse model. Our data suggest that HMGA2 inhibition offers promising perspectives for thyroid cancer treatment.

## 1. Introduction

Thyroid cancer (TC) is the most common endocrine malignant tumor, with an increasing worldwide incidence rate [1]. Papillary thyroid carcinomas (PTC) represent the most prevalent form of thyroid malignancy (80–85%). PTC is characterized by slow growth and a good prognosis. However, 10–15% of PTC dedifferentiates into more aggressive and lethal thyroid cancer with hallmarks of local invasion, distant metastases, and recurrences [2,3,4]. Genetic alterations play an important role in the initiation and progression of PTC, such as point mutations in BRAF and RAS genes and RET/PTC rearrangements, leading to the constitutive activation of the mitogen-activated protein kinase (MAPK) signalling pathway, mainly known to induce proliferation and dedifferentiation of the thyrocytes [4,5]. BRAF^V600E^ mutations account for the highest proportion of genetic lesions in PTC. They are detected in up to 50% of cases associated with aggressive clinicopathologic features, a higher rate of radioactive iodine refractory tumors, and poor outcomes [4]. Despite the knowledge gained in research on the effect of specific mutations in the development of thyroid cancer, the molecular mechanisms involved in progression, recurrences and drug resistance in TC are still unclear.

Fine-needle aspiration biopsy (FNAB) is the most widely used procedure to determine whether thyroid nodules are malignant. Nevertheless, 30% of the FNAB diagnoses are inconclusive, leading to a high rate of unnecessary surgery [4,6]. Therefore, identifying novel biomarkers is also a clinical need to improve prevention and diagnosis. In the last two decades, increasing evidence has demonstrated that microRNAs (miRNAs) play a role in the development and progression of human neoplasms, including PTC, and are among the most studied potential biomarkers in thyroid cancer [6].

MiRNAs are a major class of highly conserved small non-coding RNAs, 19 to 25 nucleotides long, which negatively regulates gene expression at the post-transcriptional level, mostly by binding to the 3′ untranslated region (3′-UTR) of their RNA target [7,8]. Because a single miRNA can modulate the expression of about 200 genes, abnormal expression can lead to the deregulation of whole signalling pathways in tumors. Our previous miRseq analysis of PTC and adjacent lymph node metastases has led to the discovery of up and downregulated miRNAs [6], among which miR-204-5p, one of the most down-regulated miRNAs [3,6,9]. This miRNA is included in miRNA molecular signatures that distinguish PTCs from normal thyroid tissues [9,10]. miR-204-5p has been shown to act as a tumor suppressor by inhibiting the proliferation of thyroid cells [11,12]. In addition, its expression is lower in PTC tumors harboring the BRAF^V600E^ mutation and in the metastases compared to the primary PTC [6], suggesting a potential inhibitory role of miR-204-5p in thyroid tumor aggressiveness. The human *MIR-204* gene is located within the intron of *TRPM3* in human chromosome 9 (9q21.12-q21.13), and both are thus under the control of the *TRPM3* promoter [13]. The mature sequence of *MIR-204* is highly conserved across mammals underlying an essential role in biological processes. miR-204-5p expression is dysregulated in various diseases, including retinopathies, arterial hypertension, diabetes, and several types of cancer [14]. While miR-204-5p has been reported to be highly expressed in insulinomas and acute lymphocytic leukemias, its downregulation is described in multiple human cancer types where it functions as a pivotal tumor suppressor [15,16,17,18] by regulating proliferation, apoptosis, stemness, chemoresistance, EMT, and metastasis in hepatocellular [19], breast [15] and prostate cancer [20]. EMT is a dynamic process that occurs during normal phases of embryonic development and wound repair [21]. It is also a complex program involved in cancer progression by promoting invasion and metastasis, during which epithelial cells lose their junctions and polarity and exhibit a molecular and morphological state of mesenchymal cells. Identifying key modulators involved in the activation of EMT in TC could also provide new therapeutic anticancer strategies.

In the present study, we aimed to study and characterize the functional role of miR-204-5p in thyroid tumorigenesis. We explored which mechanisms and pathways miR-204-5p expression is deregulated using thyroid cancer cell lines. Our data uncover an important role for miR-204-5p in the inhibition of invasion. We identified several targets of this miRNA closely associated with EMT, among which HMGA2, a well-described molecular marker able to distinguish between benign and malignant thyroid tumors [22], may be an essential contributor to tumor invasion. Finally, the clinical relevance of our results and the correlation between miR-204-5p and HMGA2 expressions were analyzed in human PTCs and a transgenic mice model of PTC. Targeting HMGA2 could therefore represent a promising therapeutic strategy for treating thyroid cancer.

## 2. Results

### 2.1. miR-204-5p Expression Is Downregulated in Thyroid Carcinomas and Thyroid Derived Cancer Cell Lines

We first analyzed miR-204-5p expression levels in normal samples, thyroid cancer tissues and metastatic samples using thyroid miR-seq data from The Cancer Genome Atlas (TCGA). The miR-204-5p expression level was significantly downregulated in primary tumors and even more in metastatic samples compared with normal samples (Figure 1a). Since the BRAF^V600E^ mutation, present in 45–80% of PTC, is associated with aggressiveness and clinicopathologic parameters of poor outcome in PTC, we compared miR-204-5p expression levels between PTC harboring the BRAF^V600E^ mutation and those carrying other mutations in the TCGA data (Figure 1b): miR-204-5p expression is significantly more reduced in BRAF^V600E^ mutated PTC compared to PTC with other mutations. Its expression is also significantly decreased according to tumor stage (Figure 1c). The downregulation of miR-204-5p was experimentally validated in 7 PTC (Table 1) compared to their adjacent non-tumorous tissue by RT-qPCR (Figure 1d), in accordance with previous results from our lab [6]. Its expression was then analyzed in three PTC-derived cell lines: TPC-1, BCPAP and K1. Figure 1e shows that miR-204-5p expression level was strongly reduced in TPC-1 and BCPAP cells and, to a lesser extent, in K1 cells compared to a pool of 8 normal human thyroid tissues. These results demonstrated that miR-204-5p is downregulated in thyroid cancer, suggesting that this miRNA could be involved in PTC progression and aggressiveness.

### 2.2. miR-204-5p Overexpression in PTC Derived Cell Lines Inhibits Their Invasion Capabilities but Has No Impact on Proliferation or Apoptosis

To investigate the biological function of miR-204-5p in PTC, we performed gain-of-function experiments by transiently transfecting miR-204-5p mimics (miR-204-5p) or mimics negative control (miR-NC) in TPC-1 and BCPAP cell lines that were chosen because they have almost completely lost endogenous hsa-miR-204-5p expression (Figure 1e). The percentage of transfection efficiency was evaluated by FACS and microscopy with a fluorescent mimic SIGLO and was around 95% for both cell lines. In both cell lines, the stability of ectopic expression of miR-204-5p was analyzed on days 1, 2, 3, 4 and 7 after transfection, and an increased expression was still observed seven days after transfection (Appendix A). Overexpression of miR-204-5p significantly decreased the number of invasive TPC-1 and BCPAP cells compared to those transfected with the negative control (miR-NC). In contrast, there was no significant effect on migration (Figure 2a). The number of EdU-labelled cells (Figure 2b) and the quantification of cleaved caspase 3 and PARP (Figure 2c) indicated that miR-204-5p had no effect on proliferation or apoptosis in TPC-1 and BCPAP cells. In addition, no effect on cell growth could be observed, as shown in Appendix A. These findings suggested that miR-204-5p could play a role in thyroid cancer cell invasion.

### 2.3. miR-204-5p Downregulates EMT-Related Genes

Next, to better characterize the molecular mechanisms underlying the effects of miR-204-5p, we measured global mRNA expression changes in TPC-1 cells three days after miR-204-5p transfection by microarray analysis, using Affymetrix Human Genome U133 Plus 2.0 arrays. mRNA expression profiles of miR-204-5p were normalized with the negative control, miR-NC. Genes more than 1.5-fold change differentially expressed (DEGs) compared to the control cells and validated by luciferase reporter assay in the literature as targets of miR-204-5p are listed in Figure 3a. Results highlighted the dysregulation of a lot of EMT-related genes like Forkhead Box C1 (FOXC1), Vimentin (VIM), the SRY-Box Transcription Factor 4 (SOX4), Snail Family Transcriptional Repressor 2 (SNAI2), Transforming Growth Factor Beta Receptor 1 (TGFBR1), Transforming Growth Factor Beta Receptor 2 (TGFBR2) and High Mobility group protein 2 (HMGA2). To help us to select interesting candidates, three online miRNA-target search tools (Targetscan, miRDB, and miRTarBase) were used to analyze predicted and validated targets of miR-204-5p globally. Only genes that were derived from the intersection of these online databases were considered eligible. Common genes between the three databases and the Affymetrix data and having a role in adhesion, motility or invasion processes are shown in red in Figure 3b.

To validate potential target genes of miR-204-5p involved in EMT and derived from the above analyses, we investigated by RT-qPCR, western blot or immunocytofluorescence whether their expression was modified in miR-204-5p transfected TPC1 and BCPAP cells: SOX4, TGFBR2, SNAI2 and HMGA2 mRNA and protein levels were decreased (Figure 4a,b). HMGA2 protein level was also significantly decreased in miR-204-5p overexpressing TPC-1 and BCPAP cells as measured by immunocytofluorescence (Figure 4c). Most results obtained by our RT-qPCR analyses validated the Affymetrix data except for HMGA2, which could be explained by cross-hybridization problems, a sub-optimal design of probes or an incorrect probe annotation [23]. Indeed, HMGA2 had previously been validated by RT-qPCR and western blot as a target of miR-204-5p in the lab, and our results thus confirmed the previous ones. These data support the role of miR-204-5p in EMT by its regulation of different EMT key players.

### 2.4. The MAPK Signalling Pathway Regulates EMT-Related Genes in PTC Cell Lines

The Ras/Raf/MEK/ERK and PI3K/AKT signalling pathways are frequently activated in papillary thyroid carcinomas [24]. To define which of those signalling pathways were involved in EMT in thyroid cells, we analyzed by RT-qPCR the expression of important EMT-related genes following 24 h of treatment with trametinib (MEKi), GDC0941 (PI3Ki), dabrafenib (BRAFi) and BMS-754807 (IGF1Ri) in TPC-1 and BCPAP cells. TPC-1 and BCPAP cells treated with DMSO were used as a negative control.

Trametinib and GDC0941 treatments were associated respectively with suppression of the RAF/RAS/MEK/ERK and PI3K/AKT signalling pathway with the almost complete loss of ERK and AKT phosphorylation in TPC-1 and BCPAP cells, as shown in Figure 5a. As expected, [25,26], the dabrafenib treatment led to an inhibition of ERK phosphorylation only in BCPAP cells that are BRAF^V600E^ positive, while it was inefficient in TPC-1 cells that are BRAF WT.

HMGA2 mRNA expression level was significantly reduced after MEKi and MEKi + BRAFi treatment in TPC-1 cells and after MEKi, BRAFi and MEKi + BRAFi treatment in BCPAP cells, while inhibition of the PI3K or the IGF1R signalling pathway had no effect (Figure 5b). HMGA2 protein expression decreased after MEK inhibition in TPC-1 and after MEK and BRAF inhibition in BCPAP cells by immunocytofluorescence. Of note, the TGFβ signalling pathway, which has been shown to play an important role in EMT, is not activated in the cell lines. Interestingly, no modulation of HMGA2 mRNA expression was observed following 1, 20 or 24 h of human recombinant TGFβ1 treatment in TPC-1 cells, while SNAI2 and SOX4 mRNA, known to be induced by TGFβ, significantly increased, suggesting that the TGFβ signalling pathway does not regulate HMGA2 expression in thyroid cells. The expression levels of SOX4, SNAI1, SNAI2, ZEB1 and CDH1 mRNA were significantly modulated after inhibition of the MAPK signalling pathway but not after inhibition of the PI3K signalling as shown in Appendix A. We also confirmed these results in WRO cells, a second BRAF^V600E^-positive thyroid cancer-derived cell line. These results suggested that the MAPK signalling pathway, but not the PI3K or the IGF1R signalling pathways, regulates the expression of these EMT-related genes in thyroid cancer cells.

### 2.5. HMGA2 Is a Target of miR-204-5p, and Its Silencing and MAPK Signalling Inhibition Decreases Invasion in TPC-1 and BCPAP Cells

Since the majority of PTC are driven by oncoproteins that activate the MAPK signalling pathway, we hypothesized that miR-204-5p could inhibit invasion of thyroid cancer cells by targeting effectors of this pathway and, more specifically, by targeting directly HMGA2, a critical regulator in cancer development that plays a pivotal role in EMT. To confirm that HMGA2 is a direct target of miR-204-5p, a luciferase reporter assay was performed: the 3′UTR of HMGA2 containing the putative miR-204-5p binding region fused to a constitutively expressed luciferase reporter was transfected into TPC1 cells. The sequence of the predicted binding site of miR-204-5p on the 3′UTR HMGA2 sequence and the corresponding mutated sequence is shown in Figure 6a. Results indicated that the luciferase activity of TPC-1 cells containing the wild type 3′UTR of HMGA2 co-transfected with mimic miR-204-5p (HMGA2-WT-204) was dramatically decreased compared to the cells transfected with a negative control mimic (HMGA2-WT-NC) (Figure 6b). No significant effect on the luciferase activity of TPC-1 cells containing the corresponding mutated sequence (HMGA2-MUT-204) following co-transfection with mimic-204-5p was detected, demonstrating that miR-204-5p can specifically directly bind to the 3′UTR of HMGA2.

To address the functional role of HMGA2, we first analyzed its expression in different thyroid cancer cell lines (TPC-1, BCPAP and K1). HMGA2 mRNA was highly expressed in the three cancer cell lines (Figure 6c). Migration, invasion, proliferation, and apoptosis assays were performed three days after the specific knockdown of HMGA2 using siRNA transfection in TPC-1 and BCPAP cells. We analyzed these processes after treatment with trametinib (MEKi) for 24 h in non-transfected TPC-1 and BCPAP cells. For both cell lines, a significant decrease in the percentage of invasion after HMGA2 knockdown or MAPK pathway inhibition was observed (Figure 6d), while no significant effect was observed on migration. Specific knockdown of HMGA2 did not impact proliferation or apoptosis. After MAPK pathway inhibition, apoptosis was not affected, while, as expected, a significant decrease in the number of EdU-labelled cells was observed (Appendix A). We confirmed that the transfection of TPC-1 and BCPAP cells with siRNA against HMGA2 was indeed associated with reducing both HMGA2 mRNA (Figure 6e) and protein (Figure 6f) expression 72 h after transfection. These results suggested the MAPK signalling pathway promotes invasion in PTC cell lines through HMGA2.

### 2.6. DNA Methylation Mediates the Silencing of miR-204-5p and Overexpression of Its Targets in PTC Cells

We further decided to investigate by which mechanism the miR-204-5p expression level is downregulated in thyroid cancer. Epigenetic silencing of tumor suppressor genes is a common molecular alteration observed during tumorigenesis, so we looked at whether promoter hypermethylation might be responsible for the downregulation of miR-204-5p. Human miR-204-5p transcripts arise from the intronic region of the gene encoding transient receptor potential melastatin-3 (*TRPM3*) [13], and their expression levels are indeed directly correlated in the TCGA thyroid cancer samples, as shown in Figure 7a (R = 0.9, *p* < 0.0001). Treatment with 5′-aza-2-deoxycytidine (5′AZA), a demethylating agent, for 24 h led to a significant increase of miR-204-5p and its host gene (TRPM3) transcripts in TPC-1 and BCPAP cells, suggesting that DNA methylation mediates miR-204-5p silencing in thyroid cancer cell lines (Figure 7b,c). We also analyzed miR-204-5p and TRPM3 transcript expression levels after trametinib (MEKi), GDC0941 (PI3Ki), and dabrafenib (BRAFi) treatment. However, we did not detect any change in expression (Figure 7b,c), suggesting that the MAPK and the PI3K signalling pathways do not directly repress miR-204-5p expression. Next, we examined if 5′AZA treatment of TPC-1 and BCPAP cells could reduce the expression of targets of miR-204-5p, and we indeed observed a reduced expression of HMGA2, SOX4, and SNAI2 mRNA while ZEB1 mRNA expression, which is not targeted by miR-204-5p, remained unchanged (Figure 7d).

### 2.7. HMGA2 Is Overexpressed in Human Papillary Thyroid Cancer

To address the in vivo relevance of our data, HMGA2 expression was first evaluated using the thyroid RNAseq data from TCGA. Figure 8a shows that HMGA2 mRNA levels are significantly increased in primary tumors and metastatic samples compared to normal samples. HMGA2 is also significantly more expressed in PTC tumors carrying the BRAF^V600E^ mutation (Figure 8b), and according to tumor stage since it is more expressed in stages III and IV compared to stage I (Figure 8c). A strong inverse correlation (R = −0.52, *p* < 0.0001) was observed between the levels of miR-204-5p and HMGA2 mRNA, as shown in Figure 8d. To validate the TCGA data, we examined HMGA2 mRNA expression in an independent cohort of PTCs and adjacent normal tissues by RT-qPCR. HMGA2 mRNA levels were strongly increased in human PTCs, while its expression was very low or absent in adjacent normal tissues (Figure 8e). Among those PTCs, only one carried the BRAF^V600E^ mutation. An inverse correlation between miR-204-5p and HMGA2 mRNA levels was also observed in our samples. Immunostainings revealed that HMGA2 protein was largely overexpressed in PTC while almost absent in normal adjacent tissues (Figure 8f). HMGA2 staining was detected in the thyroid epithelial cancer cells (KRT8 positive), with an average of 52% of these cells expressing the protein, depending on the PTC. These data highlight that HMGA2 is a promising diagnostic marker and therapeutic candidate.

### 2.8. miR-204-5p Is Downregulated While HMGA2 Is Overexpressed in the Thyroid of RET/PTC3 Mice

Cancer cell lines are only partial models of the in vivo corresponding tumors [27]. To investigate the in vivo relevance of the relationship between miR-204-5p and HMGA2 expression and their role in thyroid tumorigenesis, the use of mouse models of human cancers is very helpful. We used the RET/PTC3 transgenic mice model [28]. These mice express in their thyroid, under the control of the thyroid-specific thyroglobulin promoter, the human RET/PTC3 rearrangement, encoding a fusion protein between the RET tyrosine kinase domain and the 5′ terminal region of the ELE1 gene. This results in an abnormal expression of the RET/PTC3 oncoprotein triggering the constitutive activation of the MAPK and PI3K signalling pathways. These mice develop solid type papillary thyroid carcinoma and represent a suitable model of husolid-type9,30]. As shown in Figure 9a, miR-204-5p expression levels were strongly reduced while HMGA2 mRNA expression was enhanced in 2- and 6-month-old RET/PTC3 thyroids (RET) compared to wild type (WT) thyroids. A correlation analysis from our RT-qPCR data indicated a significant inverse correlation between the expressions of miR-204-5p and HMGA2 mRNA in 2- and 6-months old RET/PTC3 thyroids (R = −0.77, *p* = 0.0047; R = −0.71, *p* = 0.0312 respectively) (Figure 9b). HMGA2 protein expression was also increased in RET/PTC3 compared to WT thyroids, measured by western blotting (Figure 9c) as well as by immunocytofluorescence (Figure 9d). HMGA2 staining was detected in the thyroid epithelial cancer cells (KRT8 positive), with 80–90% of these cells expressing the protein. Our previous observations in human PTC are closely thus recapitulated in a mouse model of PTC and opens new research perspectives to delineate the role of miR-204-5p and HMGA2 in thyroid tumor development.

## 3. Discussion

TC is the most common endocrine malignant tumor with an increasing worldwide incidence [1]. Even though TC is usually associated with a good prognosis, there is still a clinical need for effective treatments for advanced and metastatic iodine-refractory thyroid cancers. Over the past ten years, miRNAs have received increasing attention since the discovery that their deregulation is associated with the development of diseases, especially cancer development and progression. In addition, thanks to their ease of detection, high stability, and tissue specificity, they represent excellent candidates as sensitive and specific biomarkers in the clinic [29]. Given that a single miRNA can target many mRNAs at the post-transcriptional level, it is a real challenge to fully understand molecular mechanisms and discriminate essential genes governing biological processes following the dysregulation of miRNAs in the context of cancer. In our study, we aimed to provide a better understanding of the role of miR-204-5p in PTC tumorigenesis. This miRNA was selected based on our previous study in which we had shown that miR-204-5p was one of the most downregulated miRNAs in PTC tissues and even more in metastatic samples compared to their normal adjacent tissues [6].

Moreover, its decreased expression was correlated to tall cell variant type, extrathyroidal extension, PTC metastasis, and the BRAF^V600E^ mutation, characterized by poor clinical outcome [6,30,31]. In line with our data, a bioinformatics study has suggested that the downregulation of miR-204-5p was linked to pathways and mechanisms involved in the tumorigenesis and progression of PTC [32]. miR-204-5p is also part of molecular signatures that discriminate PTCs from normal thyroids [9,10]. Consequently, our starting hypothesis was that miR-204-5p could play a key role in the progression and aggressiveness of thyroid tumors.

In the current study, using the thyroid miR-seq data collected from TCGA and a validation set of 7 independent PTCs, we confirmed that miR-204-5p expression level is strongly downregulated in PTC and that its level of downregulation is more important according to tumor stage and to the presence of the BRAF^V600E^ mutation, which is consistent with our previous and other studies [3,6,31]. A decreased expression of miR-204-5p was also observed in different PTC-derived cell lines, which were used to perform in vitro functional experiments following miR-204-5p overexpression. These functional studies have highlighted a new inhibitory role of miR-204-5p on the invasion properties of thyroid cancer cells, similar to what has already been described in other cancers, including colorectal cancer [18], oesophageal cancer [33], gastric cancer [34], prostate cancer [20] and breast cancer [35], suggesting a common role of miR-204-5p in distinct cancer types. We did not observe any effects on cell growth, proliferation, or apoptosis, contrary to what has been reported by other studies describing miR-204-5p as being involved in proliferation of thyroid cells [11,12]. These differences could be partially explained by the negative controls and cell lines used in these studies, which have distinct genetic backgrounds. Notably, Wu Z Y and colleagues have reported that miR-204-5p inhibited thyroid cancer cell proliferation in vitro. However, the cell lines used in their studies, described as thyroid cancer cell lines (KAT4 and ARO) are not of thyroid origin but are HT-29 colon cancer-derived cells ([36], cellausaurus.org, accessed on 20 April 2023). None of these studies tested the effect of miR-204-5p on migration or invasion, and proliferation was assessed using the MTT assay, which measures metabolic activity, i.e., cell viability, but does not assess de novo DNA synthesis, unlike what we did.

The molecular mechanisms involved in the inhibitory effects of miR-204-5p on thyroid cancer invasion have not been investigated so far. EMT is a significant mechanism that occurs during normal development, including implantation, embryogenesis, and tissue repair, and that can be reactivated by carcinoma cells to aid cancer progression by promoting migration and invasion of the epithelial cells. It requires cellular and molecular changes controlled by transcriptional and post-transcriptional regulatory programs involving transcription factors such as SNAI1/2, ZEB1/2, and TWIST1/2, but also by epigenetic modifications implying DNA methylation, LncRNAs or miRNAs [21]. It also involves the expression of specific cell surface proteins, the matrix metalloprotéases (MMPs), which contribute to the proteolytic digestion of extracellular matrix (ECM) constituents and then cancer invasion [21]. Our microarray analysis revealed the deregulation of many genes associated with EMT in miR-204-5p overexpressing TPC-1 cells, among which previously identified miR-204-5p targets. We validated the downregulation of SNAI2, SOX4, and TGFBR2, three targets never investigated in thyroid cancer cells, and of HMGA2, a miR-2 04-5p target previously identified in the lab (unpublished results), in TPC-1 and BCPAP cells following miR-204-5p overexpression.

Moreover, our microarray and/or RT-qPCR data revealed the downregulation of ZEB2, CDH2, ADAMTS9 and MMP14 following miR-204-5p overexpression. Our results thus suggest that the negative effect of miR-204-5p on cell invasion is mediated at least in part by the downregulation of MMP14 and ADAMTS9.

Constitutive activation of the MAPK signalling pathway is a key event in the carcinogenesis of PTC. Its overactivation can regulate many cytoplasmic and nuclear oncoproteins, impacting diverse processes such as proliferation, differentiation, migration, angiogenesis, or chromatin remodelling [37]. Interestingly, our experiments showed that inhibiting the MAPK signalling pathway decreased HMGA2, SNAI2, and ZEB1 mRNA expression, while SOX4 and SNAI1 mRNA expression increased. The increase of SNAI1 mRNA could be related to the decrease of SNAI2 mRNA through a compensatory mechanism as already described in the studies of Ying Chen et al., 2014 [38] and Lijuan Ma et al., 2021 [39]. The unexpected increased expression of SOX4 mRNA by trametinib suggests that other mechanisms are involved in regulating its expression, which is yet to be defined. It also indicates that SOX4 is not involved in the regulation of EMT by the MAPK pathway. Activation of the PI3K pathway also plays an important role in thyroid tumorigenesis. However, inhibiting this pathway did not modify the expression of the genes mentioned above. In the same way, inhibiting the IGF-1R signalling had no effect.

Another well-known potent inducer of EMT is TGFβ. Although the TGFβ signalling pathway is not endogenously activated in our cell lines, we have observed that activating this pathway by exposing the cells to TGFβ indeed strongly induced the expression of EMT transcription factors, including SNAI1, SNAI2 and SOX4 but did not modulate HMGA2 expression, indicating that HMGA2 is specifically regulated by the MAPK signalling pathway.

Assessment of the functional role of HMGA2 after knocking down HMGA2 using a siRNA revealed that the capacity of invasion of thyroid cancer cells was significantly reduced, which was also observed when the MAPK pathway was inhibited. On the contrary, we did not observe any effects on the growth, proliferation, or apoptosis of thyroid cancer cells after knocking down HMGA2, whereas, as expected, we observed a significant decrease in the proliferation rate of these cells when the MAPK signalling pathway was inhibited. This suggests that miR-204-5p could inhibit the invasion of thyroid cancer cells by targeting downstream transcripts of the MAPK signalling pathway specifically regulating invasion, including HMGA2. These data highlighted for the first time that the RAS/RAF/MEK/ERK signalling pathway may play an essential role in thyroid tumor invasion through HMGA2. Consistent with our results, HMGA2 was reportedly involved in the RAS/MEK-induced mesenchymal state in pancreatic cancer cells [40]. In addition, accumulating evidence has demonstrated the role of HMGA2 in EMT in several cancers [41], including prostate cancer [42] and colorectal cancer [43]. As mentioned in numerous studies [44,45,46,47], HMGA2 is also an essential modulator of the TGFβ signalling pathway. Therefore, one hypothesis would be that HMGA2 could induce the TGFβ pathway resulting in increasing the EMT program in thyroid cancer cells where TGFβ could, in turn, contribute to the progression of tumorigenesis through secretion of growth factors and cytokines in the stroma.

Interestingly, the study of Baquero P and colleagues showed that BRAF^V600E^ could increase TGFβ secretion in thyroid cancer cells, emphasizing the relationship between the RAS/RAF/MEK/ERK and TGFβ pathways on EMT induction [48]. However, further in vitro and in vivo studies are necessary to evaluate the potential relationship between HMGA2 and the TGFβ signalling pathway in thyroid cancer. HMGA2 is a non-histone architectural transcription factor which can bind to AT-rich sites in the minor groove of DNA, thereby affecting the transcription of numerous genes by altering chromatin structure. HMGA2 is expressed during embryogenesis and is generally not or rarely expressed in adult tissues, which makes it an interesting therapeutic target since it is re-expressed in various types of human cancer, including prostate cancer [42], breast cancer [39], oesophageal cancer [49] and thyroid cancer [50]. In addition, HMGA2 has several times been identified as a molecular marker to distinguish between benign and malignant follicular thyroid neoplasia [22,51,52]. In our study, we validated that HMGA2 is a direct downstream target of miR-204-5p by luciferase assay, in accordance with other studies [11,53]. We also showed that it is overexpressed in thyroid-derived cancer cell lines. Performing RNA sequencing following HMGA2 knockout could be interesting to analyse transcriptome changes globally. Alternatively, the CHIP-seq technology would allow identifying HMGA2 binding sites on DNA in thyroid cell lines. Moreover, to confirm our in vitro results, the assessment of the role of HMGA2 should be studied in an in vivo context.

To understand the upstream molecular mechanisms involved in miR-204-5p repression in thyroid cancer, miR-204-5p and TRMP3 expressions were evaluated after 5′-aza-2-deoxycytidine, a demethylating agent, and MEK, BRAF, PI3K or IGF-1R inhibitors treatments. Treatment with 5′-aza-2-deoxycytidine restored miR-204-5p and TRPM3 expressions, suggesting that DNA methylation epigenetically silences miR-204-5p expression, as already demonstrated in papillary thyroid carcinoma [32], in cutaneous squamous cell carcinoma [53], in glioma [54] and in leukaemia [55], indicating that DNA methylation could be a common regulatory mechanism of miR-204-5p expression in cancer.

Then, the clinical relevance of our results and the relationship between miR-204-5p and HMGA2 were evaluated in vivo in human PTC. In silico analysis of TCGA and examining independent PTC samples showed that HMGA2 mRNA and protein expression levels were strongly upregulated in PTC compared to the adjacent non-tumoral tissue. A significant inverse correlation between miR-204-5p and HMGA2 expressions was observed in our samples and the RNAseq data from TCGA. In the TCGA samples, its expression increased according to the aggressivity of the tumor, since it is more expressed in BRAF^V600E^-positive samples and higher-stage tumors.

Although cell lines are crucial to elucidate mechanisms of tumorigenesis, they are only partial models of the in vivo corresponding tumors, which constitutes a limitation of our study [27,56]. The use of in vivo mouse models of thyroid cancer will enable us to confirm and further investigate the role of miR-204-5p and HMGA2 in cell invasion. The RET/PTC3 transgenic mice model, in which the expression of the RET/PTC3 oncogene is targeted to the thyroid and which develops a solid variant of papillary thyroid carcinoma, represents a suitable model of human PTC [57,58]. These mice present a strong decrease of miR-204-5p mRNA and protein levels and, conversely, a substantial increase of HMGA2 mRNA and protein levels, at 2 and 6 months. As for human PTC, an important inverse correlation between miR-204-5p and HMGA2 expression levels was observed. In the future, these mice will be very useful to extend our in vitro observations and to characterize the roles of miR-204-5p and HMGA2 in the MAPK signalling pathway and more precisely in the control that this pathway exerts on tumor progression and invasion. In this context, the use of HMGA2 inhibitors, such as suramin, an antiparasitic drug used for the treatment of African sleeping sickness but also a potent inhibitor for HMGA2-DNA interactions [59], showing anti-cancer and anti-metastasis activities [60], is very promising.

In summary, our data suggest that miR-204-5p expression may play an essential inhibitory role in the invasion of thyroid cancer cells. Our study identified novel targets of miR-204-5p in PTC associated with epithelial-mesenchymal transition, such as SNAI2, TGFBR2, SOX4 and HMGA2. We also showed that the inhibition of cell invasion by miR-204-5p involves direct binding and repression of HMGA2, and this is specifically regulated by the MAPK signalling pathway. Finally, we confirmed in vivo the relationship between miR-204-5p and HMGA2 in human PTC and a corresponding mouse model. Taken together, our data support that HMGA2 may serve as both a potential diagnostic marker and therapeutic target in TC and suggest that HMGA2 inhibition offers promising perspectives for thyroid cancer treatment. These findings provide new insight into the molecular mechanisms underlying the aggressiveness of TC.

## 4. Materials and Methods

### 4.1. Tissue Collection

Frozen human thyroid tissues were obtained from the J. Bordet Institute. All samples were stained by hematoxylin and eosin, and their pathological status was confirmed by an anatomopathologist from the J. Bordet Institute. Protocols have been approved by the ethics committees of the institutions. Written informed consent was obtained from all participants involved in the study. The clinical information is given in Table 1.

### 4.2. Cell Lines and Treatments

The BCPAP cell line is derived from a BRAF^V600E^ positive poorly differentiated PTC and was received from Prof. G. Brabant (Department of Internal Medicine I, Lübeck, Germany). The TPC1 cell line is derived from a RET/PTC1 positive PTC and was obtained from Dr. M Mareel (University of Ghent, Belgium). The STR profile of both cell lines was also performed to ensure their purity and identity. All cell lines were maintained into RPMI1640 media (Gibco, Thermo Fisher Scientific, Merelbeke, Belgium) supplemented with 10% fetal bovine serum (FBS) at 37 °C in a humidified atmosphere containing 5% CO_2_. Cells were treated with MEK inhibitor (trametinib, 10 nM), PI3K inhibitor (GDC0941, 2.5 µM), RAF inhibitor (dabrafenib, 50 nM), TGF1R inhibitor (SB-431542, 10 µM), IGF1R inhibitor (BMS-754807, 100 nM) or demethylating agent (5′-Aza-2-deoxycytidine, 10 µM) for 24 h and then harvested for RNA or protein extraction. The effectiveness of the different treatments was confirmed by western blot analysis.

### 4.3. Transfection Experiments

Cells were transiently transfected with miRIDIAN microRNA mimic hsa-miR-204-5p (C-300563-05-0005) or negative control (#CN-001000-01-05) (Horizon Discovery, Cambridge, UK), at a concentration of 5 nM and with siRNA silencer select HMGA2 (#4392420) or negative control (#4390846) (Thermo Fisher Scientific), at a concentration of 20 nM. The transfection was performed using Lipofectamine RNAiMAX reagent (#13778150) (,Thermo Fisher Scientific) according to the manufacturer’s protocol. The overexpression of miRNA miR-204-5p and the silencing of HMGA2 were verified after each transfection by RT-qPCR analysis.

### 4.4. RNA Purification and Real-Time PCR Analysis

Total RNA, including miRNAs, was isolated from cells using the miRNeasy Mini Kit (Qiagen, Antwerpen, Belgium) according to the manufacturer’s instructions. The miRCURY LNA RT kit (Qiagen, #339340) and the Superscript III reverse transcription kit (Thermofisher Scientific, #18080093) was used for the reverse transcription of mature miRNAs and total mRNA, respectively. PCR amplification was performed using miRCURY LNA SYBR Green PCR kit (Qiagen, #339345) and KAPA SYBR FAST (Sopachem, Nazareth, Belgium) for miRNA and mRNA quantification, respectively, on ABI 7500 detection system (Biorad, Temse, Belgium). NEDD8, TTC1 and U6 snRNA was used as internal normalizers [61]. The relative expression of each gene was calculated and normalized using the 2^−ΔΔCt^ method [62]. Primers for miR-204-5p and U6 snRNA used for miRNA amplification were purchased from Qiagen (#YP00206072; #YP00203907), and the primer sequences used for mRNA amplification are listed in Appendix A.

### 4.5. Cell Migration and Invasion Assays

Cell migration and invasion were analyzed using 24-well Transwell chamber assays (8 µm pore size) (VWR, Leuven, Belgium), as described in the manufacturer’s instructions. TPC-1 and BCPAP cells were seeded into six well-plates and transfected as described above. After 48H, the medium was replaced by fresh medium without serum. The cells were harvested three days after transfection with a solution of PBS/EDTA (5 nM)/EGTA (5 nM), and 4 × 10^4^ cells per well were seeded in the upper chamber containing 500 µL of medium without serum. In contrast, the lower chamber contained 750 µL of medium with FBS as an attractant. After 20 h of incubation, the cells of the upper membrane were gently removed using a cotton swab, and the cells that had passed through were fixed and stained with azure and xanthene dyes (Polysciences, Inc, Warrington, UK). Cells were counted in 5 random fields per well using a microscope (ZOE TM fluorescent Cell imager, Biorad, Temse, Belgium). For the invasion assay, the procedures were the same except that the inserts were coated with matrigel. The percentage of invasion was calculated using the following formula: (invasive cell count/migrative cell count) × 100.

### 4.6. MTS Assay

5 × 10^3^ cells were seeded into 12-well plates, and cell viability was measured after 72 h with the CellTiter96 Aqueous Non-Radioactive Assay (Promega, Leiden, The Netherlands, #G5421), as described in the manufacturer’s instructions. Briefly, 1/20 of the MTS and 1/100 of the PMS solutions were supplemented to a fresh medium whose 700 µL were added to each well, and the plates were incubated at 37 °C in the dark for 90 min. 100 µL of the medium was transferred into 96-well plates, and the absorbance (OD value) of the formazan product at 490 nm was measured directly by a microplate reader (Biorad).

### 4.7. Proliferation Assay

According to the manufacturer’s instructions, 15 × 10^3^ cells were seeded into 6-well plates, and proliferation was analyzed 72 h after transfection using the Click-iT Plus EdU Proliferation Kit (ThermoFisher Scientific, Merelbeke, Belgium, #C10633). Briefly, the cells were incubated for 6 h with 5′-ethynyl-2′-deoxyuridine at 37 °C in the dark, and EdU incorporation was analyzed by flow cytometry on a BD LSRFortessa cell analyzer (FACS).

### 4.8. Western Blotting

For Western Blotting, lysis, and extraction of total proteins from tissues and cells were performed on ice using LAEMMLI buffer supplemented with protease and phosphatase inhibitors. Protein concentration was determined using Ionic Detergent Compatibility Reagent (IDCR) for Pierce (ThermoFisher Scientific, Asse, Belgium). Denatured proteins (30 µg) were separated by 10% sodium dodecyl sulphate-polyacrylamide gel electrophoresis (SDS-PAGE) and then transferred to nitrocellulose membranes. Primary antibodies were incubated at 4 °C overnight, and secondary HRP-conjugated goat antibodies were at room temperature for 1 h. Antibodies against HMGA2, PARP and cleaved Caspase 3 were purchased from Cell signaling Technology (Leiden, The Netherlands, #8179, #9542, #9664). Protein bands were visualized by chemiluminescence with ECL Prime Western Detection Reagent (PerkinElmer, Mechelen, Belgium, #NEL103E001EA). Protein levels were quantified and normalized to vinculin or β-actin protein levels in each sample by densitometry analysis using ImageJ software version 1.8.0.

### 4.9. Immunofluorescence

For immunofluorescence staining, tissue sections (7 µm of thickness) were fixed with 4% paraformaldehyde for 10 min and then washed two times with PBS. Subsequently, sections were blocked with a permeabilized and blocking solution containing 1% BSA 0.2% Triton X-100 5% NHS PBS for 1 h and then incubated with polyclonal rabbit antibodies against HMGA2 (Cell Signaling Technology, (Leiden, The Netherlands, # 8179S, 1:200) or cytokeratin 8 (Merck, Hoeilaart, Belgium, #TROMA-I) in a humidified chamber at 4 °C overnight. After three washes with PBS-Tween 0.05%, sections were incubated with Alexa Fluor 488 conjugated goat anti-rabbit IgG (Invitrogen, #A21206), Alexa Fluor 546 conjugated goat anti-rat IgG (Invitrogen, #A11081) and DAPI (1/30,000) for 1 h and then washed three times with PBS-Tween 0.05% and PBS. The images were scanned with a ZEISS Axio microscope, and protein levels were quantified using the QuPATH software version 0.3.2.

Affymetrix microarray hybridization and miR-204-5p target genes prediction mRNA microarray analysis was performed using Affymetrix Human Genome U133 Plus 2.0 arrays, and mRNA expression profiles of miR-204-5p transfected cells were normalized with the mRNA expression profiles of cells transfected with the negative control (miR-NC). Genes up- or down-regulated with a fold change (FC) value ≥1.5 compared to the control cells were selected as DEGs, and functional enrichment analysis was performed. To predict target genes of hsa-miR-204-5p, downregulated genes were compared to predicted (TargetScan, miRDB) and validated (miRTarBase) databases.

### 4.10. Luciferase Assay

Part of the human 3′UTR of HMGA2 mRNA sequence encompassing the miR-204-5p binding site (wild-type or mutated in the seed sequence) was cloned downstream of the Renilla luciferase gene into the psiCHECK2 vector (Promega). 1000 ng of the luciferase encoding vector was co-transfected with 50nM of hsa-miR-204-5p mimic or control in TPC-1 cells using Lipofectamine 3000 according to the manufacturer’s protocol. After 24 h of transfection, luciferase activities (Renilla and Firefly) were measured using Dual-Gluo Luciferase Assay System (Promega, #E2920) according to the manufacturer’s protocol. The relative percentage of luciferase activity was determined by the normalization of Renilla luciferase activity to Firefly luciferase activity.

### 4.11. RET/PTC3 Mice Model

RET/PTC3 transgenic mice [28] were provided by Dr. Decaussin-Petrucci (Department of Pathology, CHU Lyon, France). At 2 and 6 months, mice were sacrificed for thyroid removal. Dissected tissues were immediately placed on ice, snap-frozen in liquid nitrogen and stored at −80 °C until RNA or protein processing. All animal procedures were reviewed and approved by the university’s Animal Care and Use Committee (#CEBEA-IBMM-2020-25-86).

### 4.12. Statistical Analyses

Statistical analyses were performed using Prism GraphPad 6.0. Data distribution was analyzed by the Shapiro-Wilk normality test. The ANOVA test was used for multiple data analyses, and the *t*-test was used for data analysis between two groups. Quantitative data are represented as mean and standard deviation. All experiments were replicated at least three times independently. A *p*-value less than 0.05 was considered statistically significant.

## 5. Conclusions

In conclusion, our data provide insight into the mechanism of action of miR-204-5p in thyroid tumorigenesis, suggesting that this miRNA inhibits thyroid cancer cell invasion. Novel targets of miR-204-5p associated with epithelial-mesenchymal transition, such as SNAI2, TGFBR2 and SOX4, were identified in PTC. miR-204-5p inhibits cellular invasion notably by direct repression of HMGA2, whose expression is regulated by the MAPK signalling pathway but not by the PI3K, IGF1R or TGFβ pathways. Because therapeutic strategies based on miRNA delivery are still challenging and HMGA2 is generally not or barely expressed in adult tissues while re-expressed in thyroid cancer, HMGA2 inhibition has therapeutic potential in thyroid cancer. These findings provide new insight into the molecular mechanisms underlying thyroid cancer aggressiveness.

## Figures and Tables

**Figure 1 ijms-24-10764-f001:**
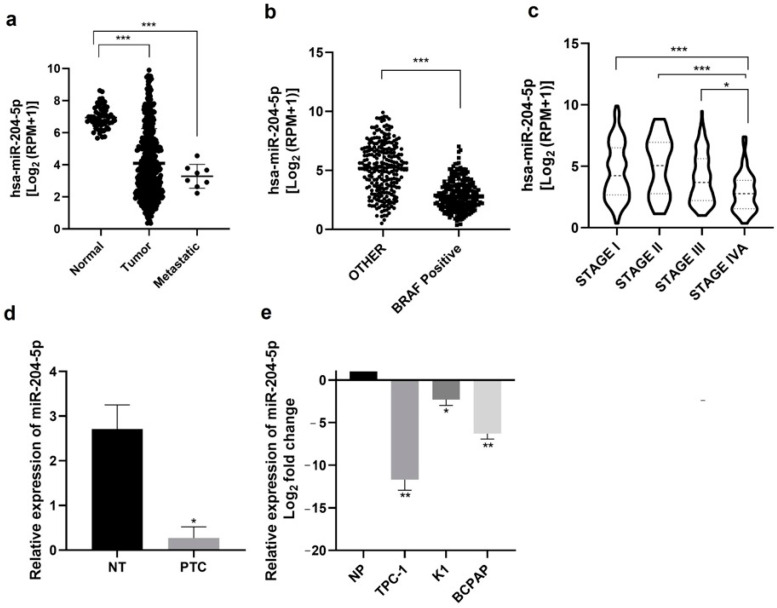
miR-204-5p is downregulated in human thyroid cancer and thyroid cancer-derived cell lines. (**a**–**c**) Analysis of miR-204-5p expression levels using thyroid miR-seq data from The Cancer Genome Atlas (TCGA), (**a**) in normal samples (n = 59), thyroid cancer tissues (n = 502) and metastatic samples (n = 8). *** *p* < 0.001 vs. normal; (**b**) in PTC tumors harboring the BRAF^V600E^ mutation (BRAF Positive, n = 212) and those carrying other types of mutations (other, n = 248), *** *p* < 0.001; (**c**) in PTC tumors according to tumor stage, * *p* < 0.05. (**d**) Analysis of miR-204-5p expression levels by RT-qPCR in PTC and adjacent normal thyroid tissues (NT) (n = 7). * *p* < 0.05. (**e**) and in PTC-derived thyroid cancer cell lines including TPC-1, K1 and BCPAP (n = 3 for each cell line) compared to a pool of 8 normal human thyroid tissues (NP). * *p* < 0.05 and ** *p* < 0.01 and *** *p* < 0.001 vs. NP.

**Figure 2 ijms-24-10764-f002:**
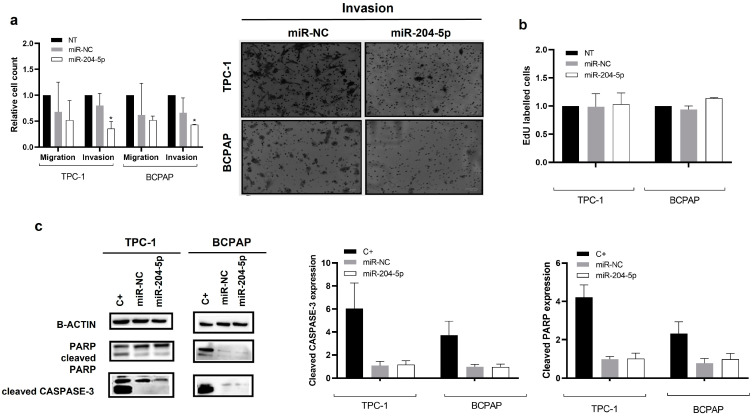
miR-204-5p overexpression inhibits invasion in PTC cell lines: (**a**) Cell migration and invasion were analyzed in TPC-1 (n = 6) and BCPAP (n = 3) cells three days after transfection, in non-transfected cells (NT), in miR-204-5p overexpressing (miR-204-5p) and mimicked negative control (miR-NC) transfected cells by counting five random fields. (**a**) Representative image after the invasion assay is shown in the right panel). * *p* < 0.05 vs. miR-NC. (**b**) EdU incorporation was analyzed in TPC-1 and BCPAP cells by flow cytometry three days after transfection in non-transfected cells (NT), and cells transfected with miR-204-5p mimic (miR-204-5p) or mimic negative control (miR-NC) cells (n = 3). (**c**) Cell apoptosis was analyzed by western blotting with cleaved CASPASE-3 and cleaved PARP antibodies and quantified by ImageJ in TPC-1 and BCPAP cells three days after transfection, in cells transfected with miR-204-5p mimic (miR-204-5p) or mimic negative control (miR-NC). TPC-1 and BCPAP cells treated with staurosporine for 20 h were used as a positive control (C+). Cleaved CASPASE-3 expression was normalized with β-ACTIN expression; for PARP, the proportion of cleaved PARP relative to total PARP (cleaved and non-cleaved) was determined.

**Figure 3 ijms-24-10764-f003:**
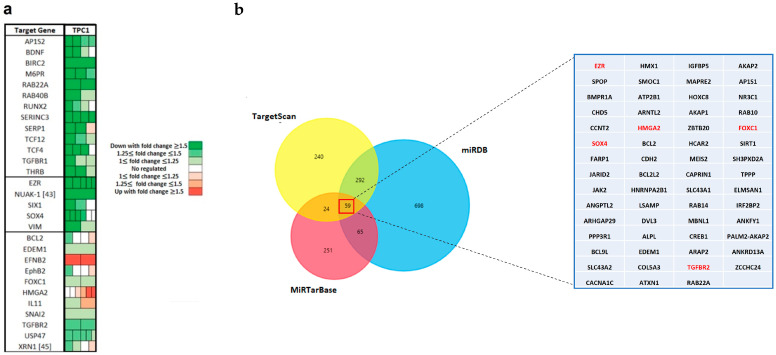
Gene expression analysis reveals a lot of EMT genes whose expression is dysregulated following miR-204-5p expression. (**a**) Heatmap of the microarray results showing the genes differentially expressed between TPC-1 cells overexpressing miR-204-5p or mimic negative control (miR-NC) and which were validated by luciferase reporter assay as being miR-204-5p targets. Fold change for each gene was defined using gene expression ratios between miR-204-5p and miR-NC transfected cells. The colors of each probe represent the amplitude of the fold changes, as shown in the legend. (**b**) Venn diagram of predicted and validated targets of miR-204-5p from three online databases. Genes common between the intersection of the three databases and the microarray results and known to be involved in cell adhesion or invasion are shown in red.

**Figure 4 ijms-24-10764-f004:**
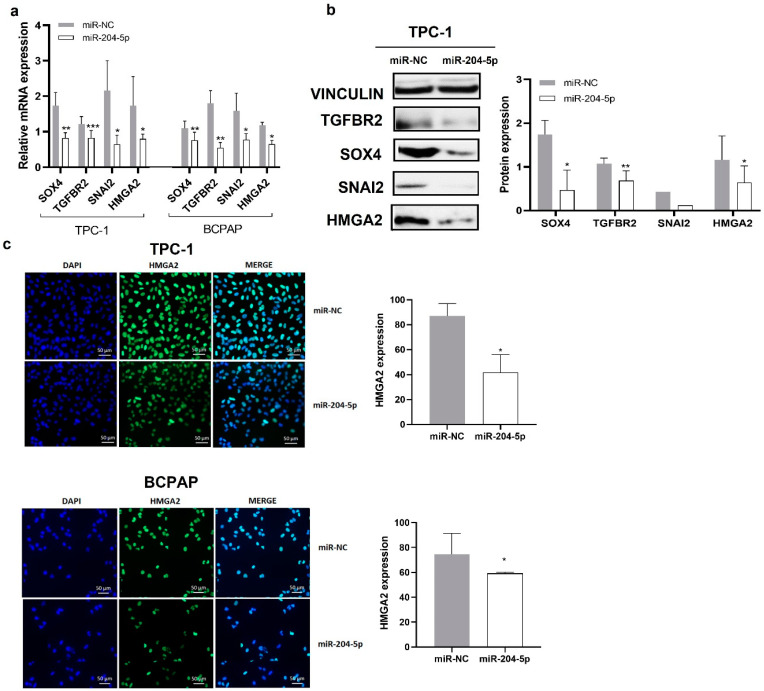
miR-204-5p inhibits SOX4, TGFBR2, SNAI2 and HMGA2 expression in TPC-1 and BCPAP cells. (**a**,**b**) TPC-1 and BCPAP cells were transfected with miR-204-5p or mimic negative control (miR-NC), and SOX4, TGFBR2, SNAI2 and HMGA2 mRNA and protein expressions were analyzed by RT-qPCR and western blotting, respectively (n ≥ 3). * *p* < 0.05, ** *p* < 0.01 and *** *p* < 0.001 (**c**) HMGA2 protein expression was also assessed by immunostaining with DAPI (blue) and HMGA2 antibody (green) 3 days after transfection. Nuclei were stained with DAPI (blue) (n = 3). The right panels represent the quantification of protein expression. * *p* < 0.05.

**Figure 5 ijms-24-10764-f005:**
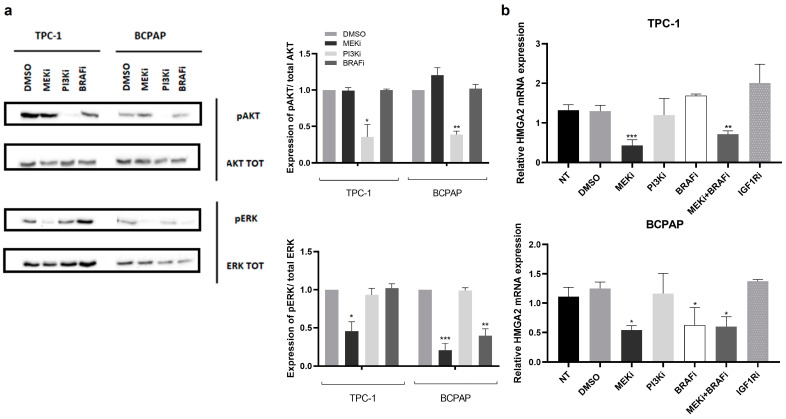
HMGA2 expression is regulated by the MAPK signalling pathway (**a**) Western blot analysis of total AKT and ERK and of the phosphorylated forms 24 h after treatment with trametinib (MEKi), GDC0941 (PI3Ki) and dabrafenib (BRAFi) in TPC-1 and BCPAP cells. Right panel: Quantification of the phosphorylated forms according to the different treatments. * *p* < 0.05 ** *p* < 0.01 and *** *p* < 0.001 vs. DMSO. (**b**) HMGA2 mRNA expression was analyzed by RT-qPCR in TPC-1 and BCPAP cells treated with different inhibitors during 24 h: trametinib (MEKi), GDC0941 (PI3Ki), dabrafenib (BRAFi), trametinib and dabrafenib (MEKi + BRAFi), BMS-754807 (IGF1Ri) (NT: non treated cells). DMSO treatment was used as a negative control. * *p* < 0.05; ** *p* < 0.01 and *** *p* < 0.001 vs. DMSO.

**Figure 6 ijms-24-10764-f006:**
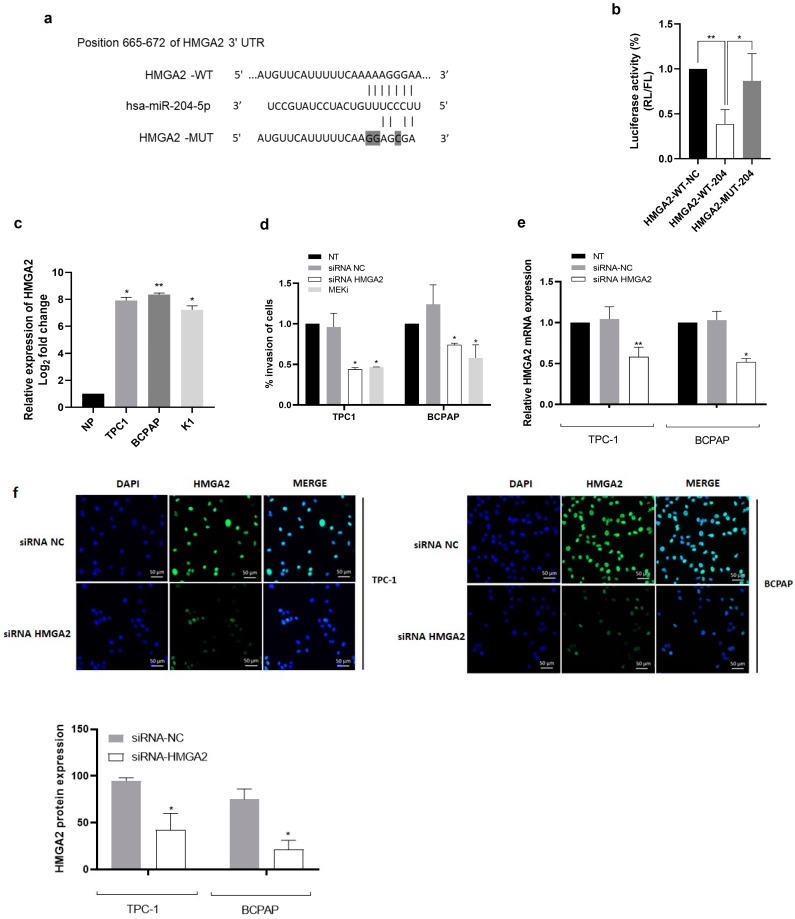
HMGA2 silencing and MAPK signalling inhibition decrease the invasion capabilities of TPC-1 and BCPAP cells. (**a**) The predicted binding site of miR-204-5p to HMGA2 3′UTR (HMGA2-WT) and corresponding mutated sequence (HMGA2-MUT). Grey content corresponds to mutated nucleotides. (**b**) Relative percentage of luciferase activity following co-transfection of TPC-1 cells with HMGA2-WT or HMGA2-MUT vector and miR-204 mimic (HMGA2-WT-204) or mimic negative control (HMGA2-WT-NC). (**c**) HMGA2 mRNA expression levels were evaluated by RT-qPCR in a pool of 8 normal human thyroid tissues (NP) and three thyroid cancer cell lines: TPC-1, BCPAP and K1 (n = 3). * *p* < 0.05 and ** *p* < 0.01 vs. NP. (**d**) Migration and invasion analysis in TPC-1 and BCPAP cells three days after transfection with a siRNA against HMGA2 (siRNA HMGA2) and a control siRNA (siRNA NC), and in non-transfected TPC-1 and BCPAP cells treated with trametinib (MEKi) for 24 h. The percentage of invasion was defined by the mean of invasion cell count divided by the mean of migration cell count x 100 (n = 3). * *p* < 0.05. (**e**,**f**) HMGA2 mRNA and protein expression analysis by RT-qPCR and immunocytofluorescence, respectively, in TPC-1 (n = 4) and BCPAP (n = 3) cells, three days after transfection with HMGA2 antibody (green) (NT = non-transfected cells). * *p* < 0.05 and ** *p* < 0.01 vs. siRNA NC.

**Figure 7 ijms-24-10764-f007:**
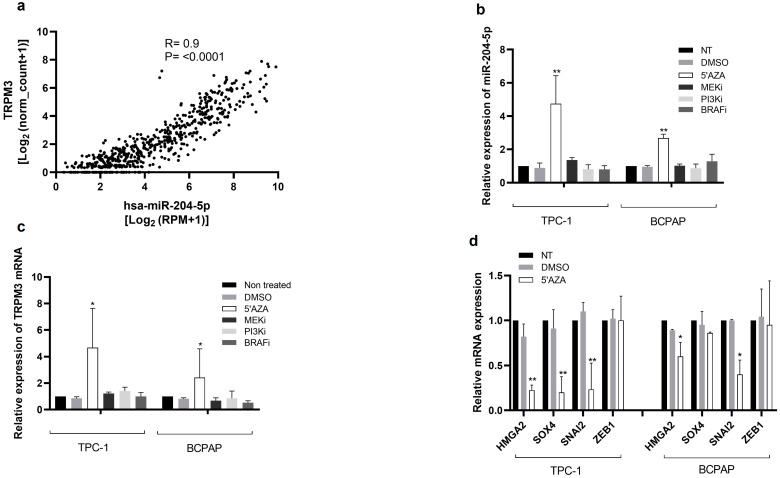
Inhibiting DNA methylation increases miR-204-5p and TRPM3 mRNA expression. (**a**) Correlation analysis between TRPM3 mRNA and miR-204-5p expression levels in the thyroid cancer samples from The Cancer Genome Atlas (TCGA). R = Pearson correlation. (**b**,**c**) Expression levels of miR-204-5p (**b**) and TRPM3 (**c**) were analyzed in TPC-1 and BCPAP cells by RT-qPCR after treatment with DMSO, 5′-aza-2-deoxycytidine (5′AZA), trametinib (MEKi), GDC0941 (PI3Ki) or dabrafenib (BRAFi) for 24 h and compared to non-treated cells (NT) (n = 5). * *p* < 0.05 and ** *p* < 0.01 vs. DMSO. (**d**) RT-qPCR analysis of mRNA expression of miR-204-5p targets (HMGA2, SOX4, SNAI2) in non-treated (NT) TPC-1 (n = 5) and BCPAP cells (n = 3) and after treatment with 5′-aza-2-deoxycytidine (5′AZA) or DMSO for 24 h. ZEB1 mRNA expression was not a miR-204-5p target and was used as a negative control. * *p* < 0.05 and ** *p* < 0.01 vs. DMSO.

**Figure 8 ijms-24-10764-f008:**
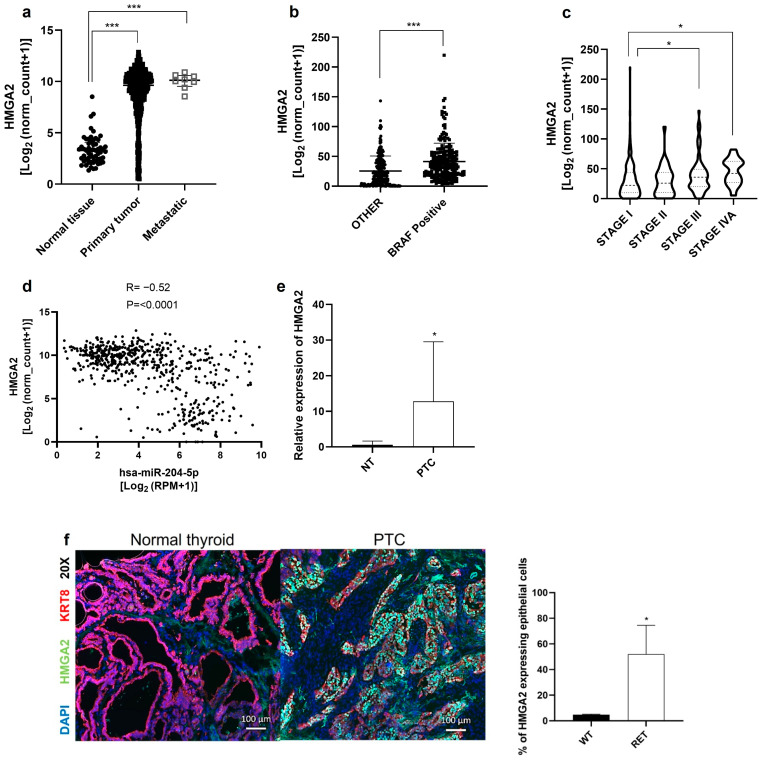
HMGA2 is overexpressed in human papillary thyroid cancer. (**a**–**d**): Analysis of HMGA2 mRNA expression using thyroid RNAseq data from The Cancer Genome Atlas (TCGA) (**a**) in normal tissues (n = 57), primary tumors (n = 502) and metastatic samples (n = 8), *** *p* < 0.001 vs. normal tissue; (**b**) in primary tumors harboring the BRAF^V600E^ mutation (BRAF Positive, n = 216) and those with other mutations (other, n = 211), *** *p* < 0.001 and (**c**) according to tumor stage. * *p* < 0.05. (**d**) Correlation analysis between HMGA2 mRNA and miR-204-5p expression in the thyroid cancer samples from The Cancer Genome Atlas (TCGA). R = Pearson correlation. (**e**) HMGA2 mRNA expression analyzed by RT-qPCR in 7 independent PTCs and normal adjacent tissues (NT). * *p* < 0.05 vs. NT. (**f**) Immunostaining of HMGA2 (green) in PTC and normal thyroid (n = 3). Epithelial cells were stained with cytokeratin 8 antibody (KRT8, red) and nuclei with DAPI (blue). The % of HMGA2 expressing epithelial cells was evaluated by the ratio: several cells positive for both HMGA2 and KRT8/number of KRT8 positive cells ×100.

**Figure 9 ijms-24-10764-f009:**
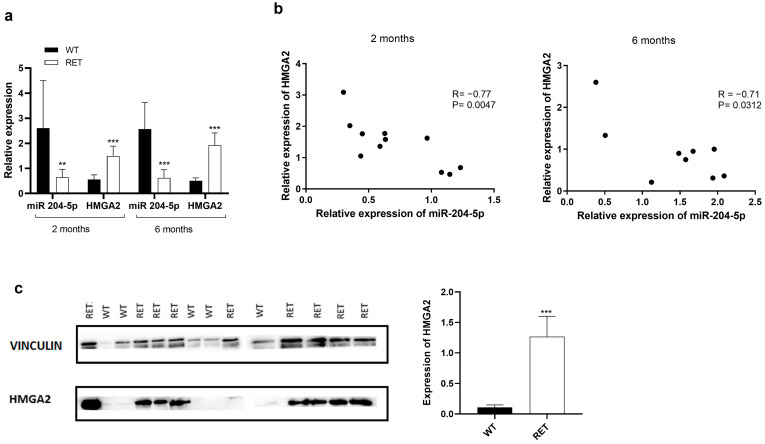
miR-204-5p is downregulated while HMGA2 is overexpressed in the thyroid of RET/PTC3 mice. (**a**) RT-qPCR analysis of miR-204-5p and HMGA2 mRNA expression in 2- and 6-month-old RET/PTC3 (RET) mice thyroids (n = 11 for two months and n = 9 for six months) and a pool of 3 normal thyroids (WT) (n = 6). ** *p* < 0.01 and *** *p* < 0.001 vs. WT. (**b**) Correlation analysis between HMGA2 mRNA and miR-204-5p expression in 2- and 6-months old RET/PTC3 thyroids. R = Pearson correlation. (**c**) Western blot analysis and quantification of HMGA2 protein expression in 2-month-old wild type (WT) and RET/PTC3 (RET) thyroids (n = 5 for WT and n = 9 for RET). *** *p* < 0.001 vs. wild-type thyroids. (**d**) Analysis of HMGA2 expression by immunostaining with HMGA2 antibody (green) and cytokeratin eight antibody (KRT8, red) and DAPI (blue, nuclei staining) in 2- and 6-month-old thyroids (n = 7). The % of HMGA2 expressing cells among the epithelial cells was defined by the ratio: several cells positive for both HMGA2 and KRT8/number of KRT8 positive cells ×100.

**Table 1 ijms-24-10764-t001:** Clinicopathological features of PTC patients: all were tested BRAF^V600E^ negative.

Patient ID	Diagnosis	Gender	Age	TNM
1	PTC, classic	F	28	pT3N1b
2	PTC, classic	F	32	pT1N1b
3	PTC, classic	F	30	pT1b(m)
4	PTC, classic	F	51	pT1N0
5	PTC, classic	F	68	pT3N0
6	PTC, follicular variant	F	23	pT2N0
7	PTC, diffuse sclerosing variant	M	14	pT3(M)N1b

## Data Availability

No new data were created or analyzed in this study. Data sharing is not applicable to this article.

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
