# Peer review of "Unraveling the Roles of miR-204-5p and HMGA2 in Papillary Thyroid Cancer Tumorigenesis"

_ijms, 2023, doi:10.3390/ijms241310764_

Round 1

Reviewer 1 Report

The study of Branteghem et al. aims to explore the role of miR-204-5p in papillary thyroid carcinoma genesis. The topic showcases a new input in the field, detailing a still underresearched subject, which could potentially lead to novel treatments in radioiodine resistant thyroid cancer. The discussions are well detailed and sustain the obtained results, and the conclusions are consistent with the evidence and arguments presented. The figures included in the manuscript are well done and accurately illustrate the results. However, I have some recommendations that I believe will enhance the manuscript.

1. I suggest adding more details about miR-204-5p in the Introduction section, such as gene location, its relation to other types of cancer (lung cancer, melanoma) etc. Phrases from the Discussion section can be moved to constitute this paragraph: lines 462-466 (“While miR-204-5p has been reported to be highly expressed in insulinomas and acute lymphocytic leukemias, its downregulation is described in multiple human cancer types where it functions as a pivotal tumor suppressor [14,30,31, 32] by regulating proliferation, apoptosis, stemness, chemoresistance, EMT, and metastasis.”), lines 553-554 (“The human MiR-204 gene is located within the sixth intron of TRPM3, and both are 553 thus under the control of the TRPM3 promoter”)

2. I do understand that this is the new manuscript format the journal requires, but I personally find it extremely hard to follow, having the Material and Methods section after the Results. I suggest adding inside the manuscript, at the very beginning of the Results section, the table describing the clinical information (provided as supplementary material), as it would make the section more intelligible and easy to follow (or at least to include a short paragraph describing the table and the specimen lot at the beginning of this section).

Other than that, the manuscript is well written, there are no detected issues on English language, and the cited references are appropriate. I believe that the manuscript can be published after minor changes.

There are no detected issues on English language.

Author Response

  1. I suggest adding more details about miR-204-5p in the Introduction section, such as gene location, its relation to other types of cancer (lung cancer, melanoma) etc. Phrases from the Discussion section can be moved to constitute this paragraph: lines 462-466 (“While miR-204-5p has been reported to be highly expressed in insulinomas and acute lymphocytic leukemias, its downregulation is described in multiple human cancer types where it functions as a pivotal tumor suppressor [14,30,31, 32] by regulating proliferation, apoptosis, stemness, chemoresistance, EMT, and metastasis.”), lines 553-554 (“The human MiR-204 gene is located within the sixth intron of TRPM3, and both are 553 thus under the control of the TRPM3 promoter”)

A1:  This has been done. We have moved parts of the discussion in the introduction and have added few more details. 

2. I do understand that this is the new manuscript format the journal requires, but I personally find it extremely hard to follow, having the Material and Methods section after the Results. I suggest adding inside the manuscript, at the very beginning of the Results section, the table describing the clinical information (provided as supplementary material), as it would make the section more intelligible and easy to follow (or at least to include a short paragraph describing the table and the specimen lot at the beginning of this section).

A2: This has been done, we have moved Table 1 from the supplementary data in the main manuscrit in chapter 2.1.

Thank you for your comments and your time. 

Reviewer 2 Report

In the manuscript” Unraveling the roles of miR-204-5p and HMGA2 in papillary 2 thyroid cancer tumorigenesis”, the authors found that miR-204-5p is downregulated with HMGFA2 upregulated in thyroid cancers. Further, the authors showed that HMGA2 expression is regulated by the MAPK pathway but not by the PI3K, IGF1R, or TGFβ pathways, and the inhibition of cell invasion by miR-204-5p involves direct binding and repression of HMGA2. HMGA2 inhibition could offer promising perspectives for thyroid cancer treatment.

In general, the study is well-designed and well-conducted. The manuscript is well-written. The language is overall clear and professional. The figures are well-made and properly labeled. The method was described with sufficient detail. 

Overall, the data and results can support the conclusions. However, there is one gap, that the authors need to take into consideration. In general, the author suggested a miR-204-5p/HMGA2 axis and MAPKs/HMGA2 in regulating thyroid cancer invasion. However, it is well-studied that MAPK pathways also regulated proliferation and apoptosis, however, according to this manuscript, miR-204-5p does not have an impact on proliferation and apoptosis. The authors need to have an in-depth discussion or have several additional pieces of evidence to have an explanation.

Here are the major points the author needs to revise or clarify:

1. In figure 2a, the authors demonstrated that miR-204-5p overexpression led to the invasion inhibition. Since the authors stated there is no migration inhibition, the authors need to check invasion-related genes, e.g., MMPs., and also EMT-related genes, e.g., Sanil1, Twist1, Twist2, Zeb1, and Zeb2, to have a mechanism-wise explanation of the invasion change.

2. In Figure 2c, to me, the cleaved PARP and cleaved CASPASE-3 both dramatically decreased in the WB image in miR-NC and miR-204-5p groups. Please double-check the WB data or the quantification.

4. In Figure 3, did the authors check other EMT-related genes, e.g., E-cad, N-cad, Snail, Zeb1/2, Twist1/2, and invasion-related genes, e.g., MMPs? Cause in Figure 2a, the author demonstrated that miR-204-5p predominantly affects invasion, but not migration, proliferation, and apoptosis. It is important to have an explanation of the invasion change with miR-204-5p overexpression.

5. In figure 4a, why miR-NC have a ~ 2-fold increase for majority of the genes? Did the author have a statistical test showing there is no significance between miR-NC and Controls?

6. In chapters 2.4 and 2.5, please use the same term to describe trametinib, either in MEKi or MAPKi, to keep it consistent.

7. In Figure 6, does MAPKi treatment and HMGA2 manipulation change the proliferation and apoptosis feature in thyroid cancer cell lines?

8. Since the author stated that Sox4, Tgfbr2, and Slug, which are all related to the stemness in cancer stem cells, were affected by miR-204-5p overexpression, I suggest the author employ a spheroid formation assay to examine the effect of miR-204-5p on cancer stem cells.

Minor:

1. In Figure 4b, why does the Snai2 have no error bar and statistical test in the bar graph?

2. In Figure 5a, is there truly no significance in pERK/total ERK in BCPAP-BRAFi? To me, it is pretty significant.

3. Figure 6 needs to be reorganized. Some information is lost in Figure 6f.

4. Please reorganize Figure 3. The font size is too small, especially the figure legends.

5. In Figure 6, are there any migration changes with HMGA2 manipulation and MAPKi treatment?

6. In the methods, the authors described that all statistical tests were on the Friedman test for multiple group experiments. Since the Friedman test is a non-parametric test, I am wondering if all the variables in the experiment are not normal distributed. Or is there a reason to not use the more widely used ANOVA test?

Author Response

Thank you for your comments and your time 
